# Macrophage Cell-Cell Interactions Promoting HIV-1 Infection

**DOI:** 10.3390/v12050492

**Published:** 2020-04-28

**Authors:** Maeva Dupont, Quentin James Sattentau

**Affiliations:** The Sir William Dunn School of Pathology, The University of Oxford, Oxford OX13RE, UK

**Keywords:** HIV-1, macrophage, T cell, virological synapse, phagocytosis, cell-cell fusion, nanotubes

## Abstract

Many pathogens infect macrophages as part of their intracellular life cycle. This is particularly true for viruses, of which HIV-1 is one of the best studied. HIV-1 infection of macrophages has important consequences for viral persistence and pathogenesis, but the mechanisms of macrophage infection remain to be fully elucidated. Despite expressing viral entry receptors, macrophages are inefficiently infected by cell-free HIV-1 virions, whereas direct cell-cell spread is more efficient. Different modes of cell-cell spread have been described, including the uptake by macrophages of infected T cells and the fusion of infected T cells with macrophages, both leading to macrophage infection. Cell-cell spread can also transmit HIV-1 between macrophages and from macrophages to T cells. Here, we describe the current state of the field concerning the cell-cell spread of HIV-1 to and from macrophages, discuss mechanisms, and highlight potential in vivo relevance.

## 1. Introduction

Macrophages are tissue-embedded cells of the innate immune system that have two distinct origins, primitive yolk sac-derived or infiltrating blood monocyte-derived cells [1]. They have essential roles in eliminating dead and dying cells, remodelling and repairing tissues, and eliminating pathogens and pathogen-infected cells. However, macrophages are also host cells for a variety of viral, bacterial, and parasitic pathogens [2]. These microorganisms must find mechanisms to enter and persist in macrophages, whilst avoiding the array of anti-microbial defenses intrinsic to these cells. Thus, the active routes of uptake by macrophages—phagocytosis, endocytosis, or macropinocytosis—usually evolve into degradative compartments—phagolysosomes, endolysosomes or autophagolysosomes [3]. Microorganisms may therefore avoid degradation by inhibiting phagosome/endosome maturation, residing in non-degradative quasi-intracellular compartments [4], or escaping into the cytoplasm [5]. A final altruistic defense is macrophage programmed cell death, either anti-inflammatory by apoptosis or pro-inflammatory by pyroptosis or necroptosis, to prevent ongoing pathogen replication and dissemination [6]. Pathogens, of which bacteria are the best studied group, have therefore evolved a sophisticated approach to evade these defences and invade, survive, proliferate in, and disseminate from, macrophages [2,5,7,8]. 

The human immunodeficiency virus type-1 (HIV-1) infects cells that express the viral receptors CD4 and one of the chemokine receptors CCR5 or CXCR4 [9]. Despite early reports of the HIV-1 infection of receptor-negative cell types, these have generally not been confirmed, with the potential exception of astrocytes that continue to be described as permissive targets for HIV-1, despite lacking conventional entry receptors [10,11]. HIV-1 strains that use CCR5 are associated with transmission between individuals and all later stages of viral infection, whereas CXCR4-using strains are more associated with later stage infection in a proportion of infected individuals [12]. T helper cells express high levels of CD4, and, once activated by cognate antigen exposure, also express CCR5, and are therefore highly susceptible to HIV-1 infection [12]. By contrast, myeloid lineage cells including monocytes, macrophages, and dendritic cells (DC) express low levels of CD4 and CCR5 or CXCR4, and are less efficiently infected by HIV-1 [12,13]. Myeloid-lineage cells also constitutively express antiviral restriction factors that potently reduce HIV-1 replication post-entry [14]. As a result, macrophages are inefficiently infectable by most viral clones in vitro, in line with the observation that so-called “macrophage-tropic” viruses, which have a higher affinity for CD4 than their non-macrophage-tropic counterparts, only arise relatively late in in vivo infection [12,13]. An extreme example of limited HIV-1 macrophage infection relates to transmitted/founder (T/F) infectious molecular clones that infect monocyte-derived macrophages (MDM) poorly or not at all, in strong contrast to primary CD4^+^ T cells that were robustly infected [15,16]. In quantitative terms, T/F viruses were on average >1000-fold more infectious for T cells than macrophages, and even so-called macrophage-tropic viral clones were ~10-fold less infectious for macrophages than T cells [15,16]. However, a caveat is that most work on macrophage infection has been carried out using cell-free virus infection. Given that macrophages are tissue-resident cells that interact with each other, and with lymphocytes in their antigen-presenting capacity, it seems logical that forms of cell-to-cell spread might be particularly relevant to in vivo viral dissemination. 

## 2. Modes of HIV-1 Cell-Cell Spread

The fluid phase diffusion of cell-free virus particles is an inefficient mode of cell infection, since virions must encounter target cells amongst a variety of non-permissive cell types, engage the cell surface, locate entry receptors, and fuse, all in the face of soluble anti-viral host defences [17].

By contrast with cell-free spread, cell-cell HIV-1 infectious spread between T cells is generally accepted to be approximately 10-fold more efficient [18,19], although higher estimates have been reported based mainly on transfer of the viral protein Gag between cells [17]. Cell-cell spread can overcome many of the obstacles to viral infection, as follows: 1. Because infected and uninfected target cells are physically engaged, virus does not have to navigate long-distance fluid phase diffusion to engage its target cell [18]. 2. Viral budding may be polarized towards the cell-cell contact site, focusing viral release towards the closely opposed target cell [20]. 3. The viral receptors cluster at the site of cell-cell contact, allowing rapid viral receptor engagement and entry [21]. 4. The high viral multiplicity of infection (MOI) imparted by cell-cell spread may assist the virus in overcoming restriction factor antagonism [22,23], and may additionally increase resistance to neutralizing antibodies against specific epitope clusters [24,25] and some, but not all, antiretroviral therapy (ART) [26,27,28]. The first mechanism of HIV-1 cell-cell spread was described for T cells, and introduced the concept of the virological synapse (VS, Figure 1) [21]. The T cell VS relied upon viral hijacking of the actin cytoskeleton, microtubule network and secretory apparatus in the infected cell to promote polarized viral budding, and of the actin cytoskeleton, adhesion molecules (e.g., ICAM-1, ICAM-3, and LFA-1) and viral receptors in the target cells to promote receptor clustering at the site of cell-cell interaction [17,29,30] (Figure 1). However, unlike the T cell VS, which is a pathogen-induced structure (since normal T cells only make very transient contacts with each other [31]), antigen presenting cell-T cell immunological synapses are part of the intrinsic function of these cells, and so might be expected to occur more readily in infected tissues. Indeed, DC-T cell clusters were first described in this regard, with highly productive T cell infection observed in coculture with HIV-1-infected DCs [32]. Two types of transfer from DCs to T cells have been described. The first is infection in cis, in which DCs and macrophages are infected by HIV-1 and release the virus to infect adherent T cells across a structure termed an “infectious synapse”. The second is infection in trans, by which HIV-1 is taken up via lectin-mediated interactions, such as by the Siglec-1 and DC-SIGN-mediated uptake [33], into a non-degradative intracellular compartment, which can then be released to infect adherent T cells [34]. This type of cell-cell in trans viral transfer event has been confirmed using intravital microscopy in mice: sinus-lining macrophages captured retroviral particles via Siglec-1 and transferred them in trans to susceptible lymphocytes [35]. Interestingly both the in cis and in trans infection routes appear to rely upon virus storage in a non-degradative compartment prior to release to infect T cells. Although not formally demonstrated, it seems likely that the virus-containing compartments (VCC) described for MDM implicated in in cis infection, and DCs in in trans infection, are the same or related membranous structures [4,36].

## 3. The Macrophage Virus-Containing Compartment

Insight into the cell biology of macrophage infection by HIV-1 has come from analysis of the compartment into which HIV-1 buds, and is subsequently released. Early electron microscopy studies observed virions accumulated in apparently intracellular compartments within macrophages [37,38], that were initially proposed to be multivesicular bodies (MVB), based upon morphology and phenotype [39]. However, subsequent analysis revealed that these vesicular structures were in fact surface-connected and at neutral pH [40,41,42], and therefore were a non-degradative compartment, within which HIV-1 could remain infectious for extended periods of time. This was proposed to be up to 7 weeks post-infection in one study [43], although this length of time might be confounded by very low level ongoing viral replication in the presence of inhibitor. Connections linking the VCC to the surface were frequently tight (~20 nm), and therefore too small for free diffusion of virions or antibodies, thereby protecting the virus from hostile elements of humoral immunity [4,44,45,46]. Live-cell imaging revealed that some VCC were open to the extra-cellular milieu, and could be accessed by small dextran particles after HIV-1 infection. However, not all VCC remained cell surface-connected, some conduits being transient and allowing or not the diffusion of small molecules in and/or out these compartments [47]. The origins of VCC remain unclear, as this structure is pleomorphic [42] and presents markers specific for the plasma membrane (e.g., CD44), but also of MVB (e.g., CD9, CD81, CD53) [39,41]. It was subsequently established that HIV-1 subverts a pre-existing compartment in macrophages to form the VCC. Following infection of MDM, viral Gag is recruited to pre-existing CD36^+^ compartments, which become VCC [42,48]. Additionally, lectins such as Siglec-1 can play an important role in VCC formation and function. A recent study demonstrated that VCC formation does not necessarily require macrophage infection, since the Siglec-1 capture of non-infectious viral-like particles (VLP) bearing HIV-Env and gangliosides also led to VCC formation [49]. Depletion of Siglec-1 from infected macrophages decreased VLP uptake, significantly reduced VCC volume, and reduced the transfer of particles to autologous T cells, confirming that VCC are important structures for viral transfer between infected macrophages and T cells [49]. The budding of HIV-1 into this type of compartment is in striking contrast to the plasma membrane budding of the virus in T cells, which allows full access of antibodies to cell surface-exposed virions. Engagement of macrophages by T cells might signal to activate actin-mediated delivery of virus from the VCC to the engaged lymphocyte. This hypothesis is supported by the observation that the viral protein Gag and tetraspanins, previously described as markers of MVB but corresponding to the VCC [39], were concentrated at the VS between infected MDM and uninfected T cells, with about 80% of the Gag staining located in the synapse [50]. As opposed to the T cell-T cell VS, the recruitment of Gag to the MDM-T cell VS was independent of Env, but could be prevented when the Pr55^Gag^ matrix protein was mutated [50]. The movement of the Gag^+^ compartment towards the VS is most likely dynamic and targeted, and was shown to rely on the microtubule network and on the kinesin KIF3A [51]. Indeed, when KIF3A was silenced by RNAi in MDM, Gag^+^ VCC accumulated in the cytoplasm, while the release of HIV-1 virions in the culture supernatant was significantly reduced, indicating that the actin/microtubule cytoskeleton is essential in directing Gag^+^ VCC towards the plasma membrane and allowing virus release, notably at the VS [51]. Interestingly, this process of enhanced T cell infection through VCC-containing HIV-1 particles may depend on VCC phenotype. Thus, when HIV-1 was within CD36^+^ VCC, the knockdown of CD36 by RNA interference or inhibition by soluble anti-CD36 antibodies prevented the release of HIV-1 particles from infected macrophages, and consequently their transfer to CD4^+^ target T cells [48]. 

Adding in vivo relevance to the idea of HIV-1-containing VCC, Ganor and colleagues reported that macrophages isolated from urethral tissue of HIV-1-infected men under cART contained HIV-1 DNA, RNA, protein, and virions in a VCC-like compartment. By contrast, viral components were undetectable in urethral T cells, highlighting the potential importance of macrophages as viral reservoirs in specific mucosal tissues [52].

## 4. CD4^+^ T Cell Infection by Spread from Macrophages

Similar to the infectious synapses described between DCs and T cells [34], macrophages can transfer high multiplicity infectious HIV-1 to T cells via transient adhesive synapses. The initial descriptions of this showed convincing transfer of infection from the infected macrophages to the primary CD4^+^ T cells [43,53,54]. Subsequent analyses revealed that the T cells adhered to the HIV-1-infected MDM, CD4 colocalized to the interface, and HIV-1 was rapidly transferred to the T cell [55]. Extended coculture of infected macrophages with the T cells led to >10% of the entire T cell culture becoming Gag^+^ within 12 h [55]. In a contemporaneous study, transfer of GFP-labelled Gag from the infected MDM to T cells was shown by live cell imaging, confirming the dynamic nature of this interaction [50]. Later live cell studies revealed how individual infected macrophages may productively infect large numbers of T cells in culture in an actin and adhesion molecule (ICAM-1-LFA-1)-dependent manner [56] (Figure 2A). 

Cell-cell spread from the MDM to T cells was not only efficient, but reduced the impact of reverse transcriptase (RT) inhibitors and some broadly neutralizing antibodies (bNAbs) on HIV-1 infection [26], similar to that observed for T cell-T cell spread [56]. Interestingly, the effect of MDM to T cell spread was highly selective for antibody inhibition, in that bNAbs targeting the gp41 membrane proximal external region (MPER) were ineffective, whereas those targeting the CD4 binding site and V3 loop-glycan epitope cluster remained potent. Reasons for this may have been at least partially steric, since Fab fragments of MPER-specific bNAbs regained neutralizing activity [56]. Impaired RT inhibitor activity appeared to be due to the high multiplicity of infection via the cell-cell route, since increasing the MOI of cell-free virus yielded similar resistance to RT inhibitors [26]. This result is in accord with the model first suggested by the Baltimore lab studying T cell-T cell spread, that high MOI infection can overcome limiting intracellular levels of RT inhibitor [27]. Additional to the highly efficient T cell infection route, MDM-mediated transfer of HIV-1 to T cells may help to seed the CD4^+^ memory latent reservoir. Upon HIV-1 infection, MDM secrete a pool of cytokines and chemokines, which attract CD4^+^ T target cells, potentially creating an ideal situation for viral dissemination [57], and DCs may assist in viral latency induction in CD4^+^ T cells during cell-cell HIV-1 transfer [58,59,60,61]. 

Macrophage phenotype is highly influenced by cytokine environment, and differentiation of MDM into “classically activated” (M1) or “alternatively activated” (M2a, M2b, M2c) MDM has profound influence on sensitivity to HIV-1 infection [62]. Moreover, macrophage phenotype has implications for function in terms of HIV-1 cell-cell spread. MDM differentiated in the presence of IL-4 (termed M-4) were found to be more sensitive to HIV-1 infection, and to subsequent dissemination of HIV-1 to T cells, than MDM differentiated in IL-13 (M-13) [63]. MDM differentiated to M1 in the presence of IFN-γ and TNF-α, or M2a in the presence of IL-4, reduced or enhanced expression of DC-SIGN respectively, which in turn modulated sensitivity to HIV-1 infection and dissemination to T cells [64], highlighting the role of DC-SIGN and related adhesion factors in HIV-1 spread. A recent analysis of macrophage phenotype in ex vivo urethral tissues [52] revealed a switch from a predominantly M1 phenotype in uninfected individuals to an intermediate (Mi) in infected, ART-suppressed individuals, implying that Mi polarization may facilitate HIV-1 reservoir formation.

## 5. Macrophage Infection by Phagocytic Uptake of HIV-1-Infected CD4^+^ T Cells.

As described above, macrophage infection by cell-free HIV-1 is inefficient, particularly for viruses that initiate infection in a new host, T/F viruses [15]. However, macrophage infection by two distinct forms of cell-cell spread has been described recently. The first was discovered when HIV-1-infected T cell lines or primary T cells were cocultured with autologous MDM. The MDM were observed by live cell immunofluorescence microscopy to engulf preferentially the infected T cells, compared to their uninfected counterparts (Figure 2B) [65]. This was then quantified by multispectral flow imaging (Imagestream™), and macrophages were observed to engulf approximately 50-fold more dying infected T cells than healthy uninfected cells [65]. Rather than elimination of the infected T cell in a phagolysosome, uptake of virally infected T cells resulted in rapid viral spread from the T cells to MDM, which became productively infected. This efficient mode of infection was true even in the context of weakly-macrophage tropic T/F virus clones [65]. However, despite the robust nature of infection resulting from transfer of the virus from infected T cells, this did not overcome intrinsic tropic restrictions, since MDM engulfment of X4 “T cell-tropic” virus-infected T cells did not infect the MDM [65]. Thus, virus infection of MDM used the conventional route of CD4 and CCR5-mediated entry [65]. The “eat-me” signals for MDM exposed on HIV-1-infected T cells have yet to be elucidated, but do not appear to be the more common signals associated with apoptotic cell uptake by phagocytes, as a panel of efferocytosis inhibitors failed to significantly reduce HIV-1-infected T cell uptake by MDM [65]. This was the first description of a “phagocytic” VS, and joins the small number of other phagocytic structures implicated in pathogen cell-cell spread, including bacteria [66,67] and parasites [68], and may therefore represent a general mode of pathogen dissemination [69].

## 6. Potential Consequences of Phagocytic Uptake of HIV-1 Infected Cells

All modes of cell-cell infection described above rely upon viral fusion with the target cell using CD4 and a chemokine receptor, regardless of the virus producer or the target cell type. It was therefore puzzling that a large body of literature describes human astrocyte infection, despite the fact that these cells do not appear to express CD4 or CCR5, at least in culture [70,71,72,73]. Equally difficult to explain is the well-documented finding that transfection or transduction of the HIV-1 genome into astrocytes, bypassing the limiting entry step, leads to productive viral replication in astrocytes [74,75,76]. How does HIV-1 infect cells that lack conventional entry receptors, and then fail to yield a productive infection in otherwise permissive cells? One explanation for this may come from the observation that astrocytes are endocytic and phagocytic and are capable of internalizing virions and engulfing cell debris from their surroundings into a non-degrading compartment [77,78,79]. Using sensitive viral entry assays, we confirmed the absence of HIV-1 fusion with astrocytes, and then went on to show that astrocytes make intimate contacts with HIV-1-infected MDM, and engulf virus-containing cell debris from these cells [70]. This leads to positive signals for viral infection in cells that are otherwise not productively infected, since the astrocytes will contain viral proteins and nucleic acids. Whilst these results do not formally exclude low-level infection of a small percentage of astrocytes by an unconventional entry mechanism, they do provide an explanation additional to the endocytic capacity of astrocytes [80] for the apparent observation of infected astrocytes, when exposed to the virus or virally-infected cells in vitro and in vivo [81,82]. A similar explanation may also apply to the observations that a small percentage of monocytes isolated from HIV-1-infected individuals contain nucleic acid signals for both HIV-1 and the T cell receptor [83]. Since monocytes are considered to be relatively resistant to HIV-1 infection as a result of high levels of restriction factor expression, their ability to phagocytose HIV-1-infected cells and their debris may contribute to false positive infection signals [84].

## 7. Fusion between HIV-1-Infected CD4^+^ T Cells and Macrophages

HIV-1-induced fusion of T cells with macrophages has been evoked as a mode of viral spread between these cell types that may occur in parallel with VS-mediated viral spread [85], similar to the early work by the Steinman lab on DC-T cell syncytium formation driven by HIV-1 infection [32,86]. Env-mediated fusion between monocytic and T cell lines revealed that the ensuing heterokaryons were viable, stable, and presented a dominant activated monocyte-like phenotype [87]. Revisiting the concept of myeloid-T cell fusion more recently [88], coculture between acutely HIV-1-infected activated CD4^+^ T cells and MDM culminated in the rapid appearance of Gag^+^ multinucleated MDM that expressed phenotypic T cell markers (CD2, CD3, Lck), and which was prevented by HIV-1 fusion inhibitors. Infected MDM subsequently fused with adjacent MDM, and the resulting strongly Gag^+^ multinucleate heterokaryons were viable, and released the virus for at least one month [88]. The authors did not observe obvious phagocytic uptake of HIV-1-infected T cells, and proposed the explanation that because the T cells were infected for only 3 days, T cells did not expose efferocytic “eat-me” signals, but did express high levels of the plasma membrane Env, mediating cell-cell fusion. An additional aspect of the formation of MDM-T cell heterokaryons is that the incorporation of T cell plasma membrane into the MDM appeared to allow rapid budding of virus at the heterokaryon plasma membrane, probably accounting for the subsequent MDM-MDM fusion. This is in contrast to the normal budding profile of macrophages into VCC-like compartments, which would presumably prevent MDM-MDM fusion [4,39,89], and suggests that T cell derived PIP_2_ might remain located at the heterokaryon plasma membrane, transiently refocusing viral budding to this site. The infected T cell fusion observation was extended to immature DCs and osteoclasts, implying a general mechanism of HIV-1 dissemination between T cells and myeloid-lineage cells [90]. Intriguingly, the heterokaryons formed between MDM at least partially overcame the restriction imposed on myeloid cells by SAMHD1 [90]. It was suggested that merging of the T cell cytoplasm with that of the MDM would transfer cyclin/CDK components and dNTPs that might inactivate/saturate SAMHD1 activity, and/or that the T cell nucleus containing integrated provirus may continue to transcribe viral DNA within the heterokaryon, bypassing the reverse transcription antagonism imparted by SAMHD1 [90]. Since multinucleated macrophages have been described in vivo, it seems likely that this process has biological significance, particularly in the brain where microglia and perivascular macrophages are infected [91], and in immune tissues where myeloid and T cells are in close proximity such as lymph nodes, tonsils, spleen, and gut-associated lymphoid tissue. 

## 8. HIV-1 Spread between Macrophages: Implication of Tunneling Nanotubes.

Tunneling nanotubes (TNT) are a novel type of cell-to-cell communication machinery, allowing distant cells from different lineage—for example immune cells and neuronal cells [92]—to exchange information about their environment. They are F-actin rich structures connecting at least two cells without touching the substrate in 2-dimensional cultures in vitro, which allow the transport of calcium flux, genetic material, proteins, and organelles, including lysosomes and mitochondria, between distant cells, up to 200 µm apart [93,94,95]. Interestingly, TNT can be either open-ended, allowing for the direct transfer of cytoplasm content, or close-ended and conjugated with a Gap-junction, which allows them to fulfil their transfer function [95,96]. In macrophages, thin TNT (often short, with a diameter of less than 0.7 µm) that contain only F-actin can be distinguished from thick TNT (or long, with a diameter above 0.7 µm), which comprise both F-actin and microtubules [93]. These structures are hijacked by pathogens, specifically HIV-1, which triggers TNT formation in human MDM without altering their length [97]. Eugenin and colleagues proposed that short TNT could transport single HIV-1 particles, while long TNT may carry larger HIV-1-containing vesicles (e.g., exosomes or VCC), either within or on the outside of the nanotube [97]. However, endosome, Golgi and endoplasmic reticulum proteins (ER) components were found in the proteome of TNT, and were associated with HIV-1 Gag and Env proteins, suggesting that endocytosed HIV-1 was transported to the ER and Golgi apparatus before being transported into TNT [98,99]. In addition, HIV-1-related Rab11^+^ endosome compartments were found to traffic through TNT between human MDM, in a myosin-II and actin-dependent manner [98]. In a recent study, connexin 43, a component of the gap junction, was found to accumulate in intracellular compartments that localized both at the base and tip of the TNT formed from the infected cell [100]. Importantly, HIV-1-infected MDM formed TNT that allowed gap junction communication between distant cells. Blocking these gap junctions resulted in inhibition of TNT formation and decreased HIV-1 spread in culture [100], supporting the important roles of both TNT and the gap junctions in HIV-1 cell-to-cell transfer. Viral proteins play a critical role in modulating TNT formation. For example, the HIV-1 accessory protein Nef has been recognized as a critical inducer of HIV-1-mediated TNT, through its interaction with the ubiquitous M-sec protein, an important molecular factor in TNT and cell protrusion biology [101]. HIV-1-associated TNT were also described between T cells [96]; however, macrophages and T cells TNT display different properties: in T cells, HIV-1 uses pre-existing TNT and is unable to trigger TNT formation [96]. This appears to be due to absence of M-sec expression in T cells, whereas MDM TNT were induced by Nef interactions with M-sec [101] and with myosin-X [102]. M-sec association with protein members of the exocyst complex are part of the TNT assembly mechanism [103]. Nef was found to interact with five components of the exocyst complex in Jurkat T cells [104], and depletion of one of these interactions led to a reduction in the number of TNT [105]. Heterotypic TNT formation between MDM and T cells can occur, and favours viral spread among different cell types [106]. It was described that Nef alone was transported from macrophages to T cells via TNT, a process that resulted in a decrease of CD4 expression in recipient T cells [102], and thus limiting cell-free infection events in these cells. Taken together, these studies indicate that HIV-1, mainly through Nef-dependent interactions with host proteins, hijacks physiological nanotube structures to ensure its spread and evade detection by the immune system. 

## 9. Implication of Co-Infections in HIV-1 Intercellular Macrophage Interactions

HIV-1 infection is associated with comorbidities that tend to elevate morbidity and mortality. Among these comorbidities, co-infection with *Plasmodium falciparum*, the etiological agent for Malaria, results in enhanced infection and accelerated progression to AIDS. A mechanism that may explain greater viral load is the increased HIV-1 infection of target cells. The treatment of immature DC with the malarial pigment hemozin was reported to drive rapid DC maturation, known to be more susceptible to HIV-1 infection than their immature counterparts [107]. Pre-exposure of DC to hemozin promoted HIV-1 transfer to CD4^+^ T cells, while protecting DC from productive infection with the virus [108]. Another pathogen that adversely impacts HIV-1 pathogenesis is co-infection with *Mycobacterium tuberculosis* (Mtb), the causative agent of Tuberculosis. Co-infected patients display a rise in viral load, both in the bloodstream and at the anatomical sites of co-infection [109,110,111], and this phenomenon was associated with macrophage rather than T cell infection [112]. Amongst the mechanisms proposed to explain this phenomenon, the release of TNFα, IL-6, IL-1β and IL-10 by Mtb-infected MDM favoured the viral replication in HIV-1 infected cells, by increasing NFκB binding to HIV-1 LTR sequences [113,114]. Recently, we reported that the Mtb-associated microenvironment induced both thin and thick TNT formation between MDM in an IL-10/STAT3-dependent manner. This increased TNT formation was largely responsible for the enhanced viral replication and dissemination in the culture, since the pharmacological inhibition of these structures reversed the infection levels to that of control cells [115]. We also identified Siglec-1, previously described as important in VCC formation [49] and in the capture and transfer of HIV-1 from infected DC and MDM to CD4^+^ T cells [116], to be upregulated both in MDM differentiated in an Mtb-associated microenvironment, and in lung macrophages of Mtb and Mtb-SIV co-infected macaques [117]. Interestingly, Siglec-1 was highly distributed on long and thick TNT, which correlated with the viral content of these structure (Figure 3). By silencing Siglec-1, we showed that it was required for the TNT-mediated spread of HIV-1 among MDM [117]. Interestingly, Siglec-1 is not the only relevant HIV-1 receptor whose expression is enhanced by Mtb infection. Other HIV-1 adsorption receptors upregulated by Mtb include lectins involved in HIV-1 capture [118], such as mannose receptor [119,120], and entry receptors CD4, CCR5, and CXCR4 [121]. These receptors may localize both to the plasma membrane and to TNT, thus enhancing HIV-1 capture and transfer between distantly connected cells. 

## 10. In vivo Evidence for HIV-1 Cell-Cell Spread in Macrophage Infection and Dissemination

Despite the description of several mechanisms of cell-to-cell transfer allowing HIV-1 spread, all of them have been described using in vitro systems. Yet it is likely that some of them also occur in vivo. Some evidence indicates that the formation of VS could occur in vivo, and possibly result in target cell infection through one or more of the mechanisms described above. Using intravital microscopy of HIV-1-infected humanized mouse lymph nodes, the group of Mempel found that about 10% to 20% of infected T cells presented with elongated and thin membrane protrusion of sometimes more than 100 µm in length, suggesting the formation of TNT-like structures in vivo [122]. These elongated structures appear to represent multinucleate syncytia arising from Env-mediated interactions between circulating and arrested HIV-1-infected T cells, suggesting that the first stage of viral dissemination may occur via TNT formation followed by the establishment of a VS. Additionally, productively infected T cells efficiently migrated to the lymph node cortex, microns away from the subcapsular sinus (SCS), enhancing the viral spread in the tissue [122]. The role of macrophages in this process has recently been studied in mice infected with murine leukaemia virus (MLV) or HIV-1. Upon foot-pad injection of fluorescently labelled viruses, Sewald and colleagues found that both MLV and HIV-1 accumulated in the floor of the SCS, and were associated with Siglec-1^+^ cells, most of which were CD11b^+^ macrophages [35]. Interestingly, the inhibition or knock-out of Siglec-1 abolished the viral uptake in the peripheral lymph node, testifying to the importance of macrophages in the early phase of virus uptake and spread [35]. Indeed, the authors found that MLV-GFP viruses were selectively transferred from Siglec-1^+^ macrophages to a subset of B cells in the lymph node after formation of synaptic contacts, indicating that macrophages not only captured the virus and stored it in VCC, but also transferred it to target cells through VS that recruited the VCC to the contact zone [35]. Moreover, the VS were characterized, and cell protrusion containing viral particles, along with intracellular vesicles and mitochondria, were observed, indicating the formation of TNT between infected and uninfected cells in the tissue. 

Non-human primate (NHP) studies have provided further evidence supporting the mechanism by which macrophages become infected through phagocytosis of, or fusion with, infected T cells. Indeed, SIV DNA has been detected in both CD4^+^ T cells and myeloid cells isolated from mucosal and lymphoid tissues in infected Asian macaques [123]. In SIV-DNA^+^ myeloid cells, present in lymphoid tissues where CD4^+^ T cells persist, rearranged TCR-γδ DNA was also found, indicating that these cells probably acquired viral material through infected T cell phagocytosis or fusion [123]. Furthermore, SIV-DNA-containing tissue macrophages contained replication competent viruses, similar to those found in CD4^+^ T cells in untreated animals, whereas SIV DNA levels were decreased and no replication-competent virus could be detected in tissue macrophages from ART-treated animals [124]. These results were further supported by the detection of TCR-γδ DNA in alveolar macrophages isolated from eight out of nine HIV-infected patients under ART for 3 years, indicating that T cell phagocytosis may occur in HIV patients, and could therefore participate in viral dissemination [124]. In another study, the histological section of the human lymph node and tonsils isolated from HIV-infected patients were used to determine the cell types infected in vivo. Infected CD4^+^ T cells were mainly located in the germinal centre and paracortex of the tissue, while the infected macrophages were in close proximity to the lumen of paracortical blood vessels. In this area, macrophages not only contained p24-positive material, but also infected T cells, along with p24-positive debris within phagosomes, supporting the hypothesis that in secondary lymphoid organs, macrophages can become infected by the phagocytosis of infected T cells [125]. 

## 11. Discussion

The disseminated, ubiquitous tissue location and ability of macrophages to be infected by HIV-1 and then pass the virus to other cell types makes them a cellular “infectious hub” that may remain infected for extended periods and seed virus to other macrophages and to T cells via diverse mechanisms (Figure 4). The mechanisms by which HIV-1 infects macrophages and is then transmitted onwards to other cell types in vivo is unclear, but none of the proposed mechanisms (phagocytosis, fusion, nanotubes) is mutually exclusive, and so these may function in parallel. A limitation of tissue culture models of macrophage cell-cell infection is that they almost all use MDM, whereas, in vivo, tissue-resident macrophages may be derived from early embryogenesis or blood monocyte infiltration and differentiation. Extracting macrophages from tissues is complex and will alter their phenotype, as this is critically-dependent on the local cell and cytokine environment [126,127,128]. An alternative approach that is gaining traction is the differentiation of macrophages from induced pluripotent stem cells in a selective cytokine environment, a process that may allow more physiologically-relevant cell phenotypes to be probed in HIV-1 cell-cell infection systems [129]. Similarly, almost all in vitro analyses have used plastic as the solid phase for macrophage differentiation, coculture with other immune cells and viral cell-cell infection studies. However, macrophages exist in a 3-dimensional tissue matrix that can be better mimicked using semi-solid matrices, and macrophage behavior including migration is best explored in this setting [130,131]. This, together with ever more sophisticated intravital microscopic analyses [132], will lead to a more coherent and integrated understanding of how HIV-1 moves between myeloid and lymphoid cell targets and the implications of this for prophylactic, therapeutic and cure strategies.

## Figures and Tables

**Figure 1 viruses-12-00492-f001:**
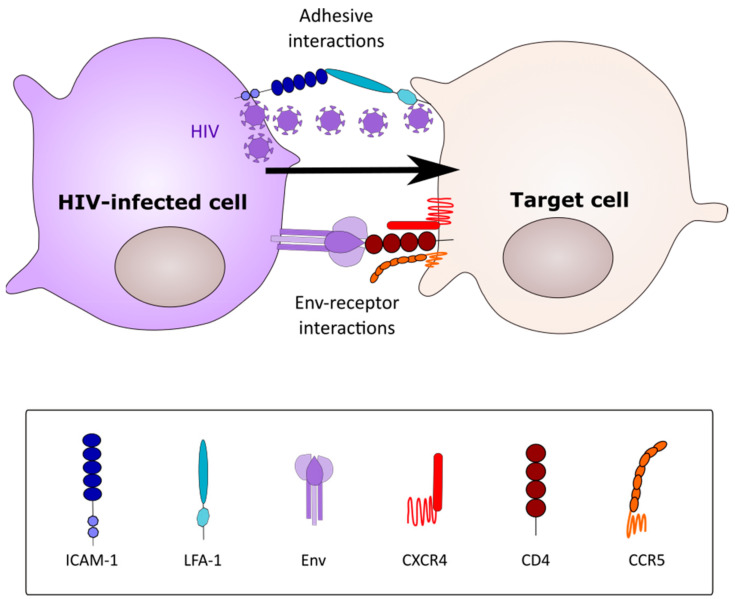
A prototypical virological synapse. The infected cell (CD4^+^ T cell or macrophage) forms a transient adhesive contact with a receptor-expressing target cell via Env-CD4 interactions (CD4^+^ T cell-T cell synapses) or via adhesion molecule interactions (macrophage-CD4^+^ T cell synapse). Virus buds from the infected cell and migrates towards receptors clustered on the target cells (arrow). Infection follows via conventional receptor-mediated entry.

**Figure 2 viruses-12-00492-f002:**
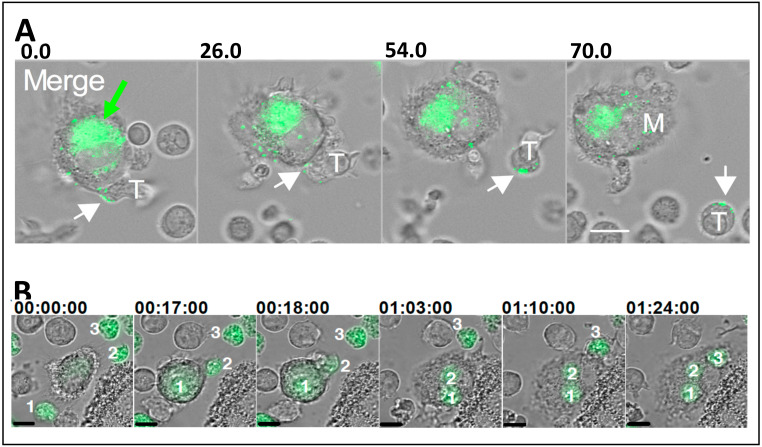
HIV-1 transfer between macrophages and CD4^+^ T cells. (**A**) HIV-1 infected macrophages transfer GFP-HIV-1 to a CD4+ T cell that is scanning the macrophage surface via transient adhesive interactions (taken from [56]). The green arrow indicates the virus-containing compartment. The white arrows show viral particles engaged with the T cell surface. Numbers above images represent minutes:seconds after initiating coculture. Scale bar = 10 μm. (**B**) HIV-1-GFP infected CD4^+^ T cells being engulfed by a macrophage (taken from [65]). Numbers above images represent hours:minutes:seconds. Scale bar = 10 μm.

**Figure 3 viruses-12-00492-f003:**
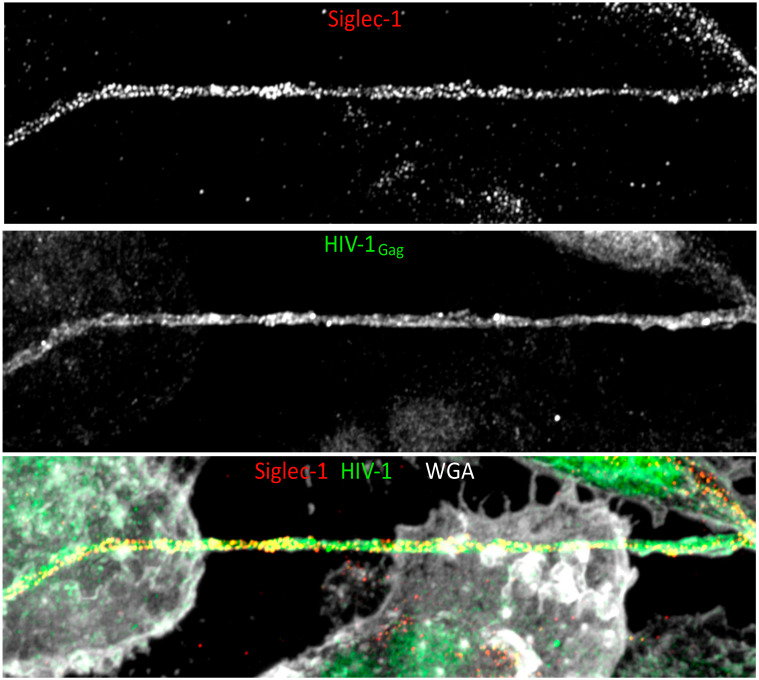
HIV-1 spreads between macrophages through Siglec-1^+^ tunneling nanotubes (TNT). Primary human monocytes were differentiated for 3 days with supernatant from *Mycobacterium tuberculosis*-infected MDM, and then infected with HIV-1_NLAD8_-VSVg. Representative immunofluorescence labeling showing HIV^+^-Siglec-1^+^ thick TNT (taken from [117] and used under CCBY 4.0). Staining shows extracellular Siglec-1 (top) intracellular HIV-1_Gag_ (middle), and cell plasma membrane stained with Wheat Germ Agglutinin (WGA, grey) with all 3 colors merged in the lower image. Scale bar: 10 μm.

**Figure 4 viruses-12-00492-f004:**
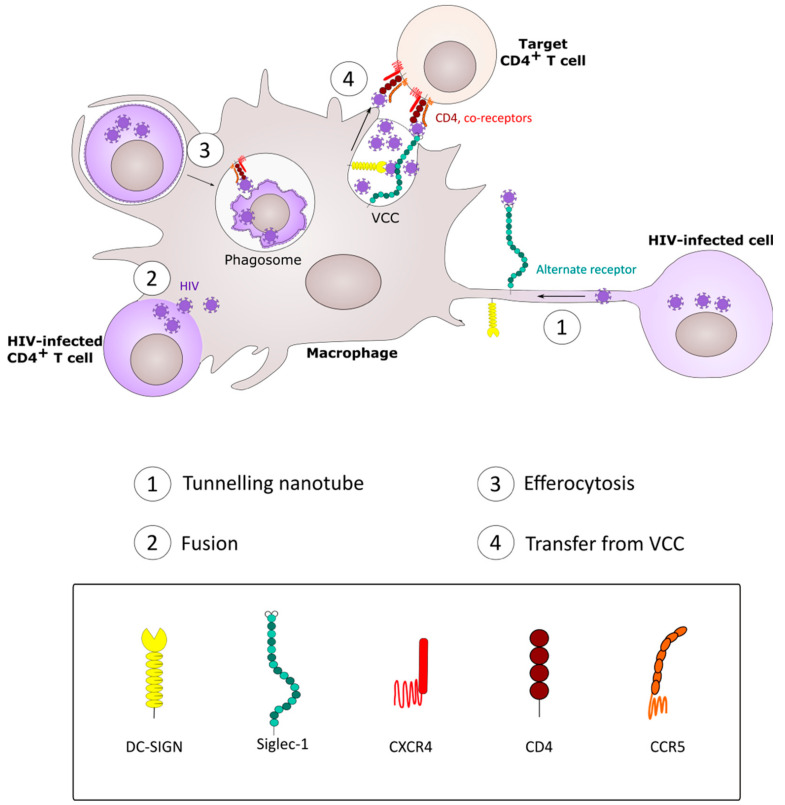
Model for role of the macrophage in cell-cell HIV-1 dissemination. Macrophages can become infected through different mechanisms, and participate in HIV-1 dissemination within the host. (**1**) HIV-infected cells form intercellular bridges—TNT—structures hijacked by the virus to spread from one cell to another, trafficking on the inside or outside of the structure, possibly by selective use of attachment receptors (e.g., Siglec-1, DC-SIGN). (**2**) Macrophages fuse with acutely infected CD4^+^ T cells, leading to a viable, infected heterokaryon. (**3**) As professional phagocytes, macrophages recognize infected and dying CD4^+^ T cells. Release of viral particles from the T cell during capture and phagocytosis allows high multiplicity macrophage infection. (**4**) In infected macrophages, the virus buds into a specialized virus-containing compartment (VCC). Upon encounter with CD4^+^ target T cells, VCC containing either free virions or attachment receptor-bound viral particles release virions that infect the target cell.

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
