# Peer review of "Macrophage Cell-Cell Interactions Promoting HIV-1 Infection"

_viruses, 2020, doi:10.3390/v12050492_

Round 1

Reviewer 1 Report

In this manuscript Dupont and Sattentau provide an overview on how macrophages serve as a critical vehicle for HIV spread in the host during infection. The authors review the literature on the different modes of macrophage infection by HIV and how these cells serve as a central axis for subsequent infection of CD4 T cells and other macrophages and myeloid cells. The manuscript provides an updated review on this topic and can thus be useful for researches in this or other fields to get acquainted with the literature.

A few comments that in my opinion could serve to improve the manuscript:

The authors insist on several occasions on the fact that in vitro macrophages are poorly infected by cell free virions. However, they do not quote any quantitative data nor study to support this conclusion. As far as I know, a careful and direct comparison of the infection rates of activated T cells and macrophages from the same donors exposed to the same viral preparations at different MOIs is still missing. If such a study exists, it should be quoted. If it does not, it might be worth to be mentioned.

The authors should include the recent findings by Ganor and colleagues (Nature Microbiology, PMID: 30718846) reporting on the persistence of infected macrophages in urethral macrophages from patients under ART. These also displayed VCCs, further supporting a role for these structures in viral spreading. In this context, the authors could speculate a possible role for residual viral transmission via macrophages as a mechanism contributing for preservation of the viral reservoir under ART. 

In the section devoted to the VCC and its connection to the plasma membrane, the authors should quote here two additional studies. In the first one, 10.1371/journal.pone.0069450, evidence supporting immediate access of the VCC to small dextran present in the extracellular medium was presented. In addition, the data presented revealed the dynamic nature of the VCC connection to the plasma membrane. In the second one, 10.1084/jem.20130566, time lapse microscopy of the same macrophages before and after infection demonstrated that HIV-1 can subvert preexisting CD36+ compartments and convert them into VCCs.

Minor issues:

Line 101: While the authors refer that HIV virions in VCCs remain infectious for only a few days, the reference they cite refers a period of 6 weeks within which infectious virions could be recovered. They authors could here point to the limitation of the measurement of infectious viral particles performed in this study.

It would be interesting to include the findings by Hammonds and colleagues (Plos Pathogens, 2017, PMID 28129379) when discussing in cis and in trans infection, or, alternatively, in the VCC section. This work reports that productive infection of macrophages is not required for formation of the VCC and that virions that result from productive macrophage infection can intermingle with virions captured from the extracellular space via SIGLEC-1.

Author Response

Comments and Suggestions for Authors – REVIEWER 1

Author replies in italics for clarity

In this manuscript Dupont and Sattentau provide an overview on how macrophages serve as a critical vehicle for HIV spread in the host during infection. The authors review the literature on the different modes of macrophage infection by HIV and how these cells serve as a central axis for subsequent infection of CD4 T cells and other macrophages and myeloid cells. The manuscript provides an updated review on this topic and can thus be useful for researches in this or other fields to get acquainted with the literature.

We thank the reviewer for their positive consideration of the manuscript and address their comments below.

A few comments that in my opinion could serve to improve the manuscript:

The authors insist on several occasions on the fact that in vitro macrophages are poorly infected by cell free virions. However, they do not quote any quantitative data nor study to support this conclusion. As far as I know, a careful and direct comparison of the infection rates of activated T cells and macrophages from the same donors exposed to the same viral preparations at different MOIs is still missing. If such a study exists, it should be quoted. If it does not, it might be worth to be mentioned.

This is a good point and we have now added a paragraph and references that address the quantitative differences between MDM and CD4+ T cell infection by HIV-1 infectious molecular clones as follows (line 53): “An extreme example of limited HIV-1 macrophage infection relates to transmitted/founder (T/F) infectious molecular clones that infect monocyte-derived macrophages (MDM) very weakly or not at all, in strong contrast to primary CD4+ T cells that were robustly infected [15, 16]. In quantitative terms, T/F viruses were on average >1000-fold more infectious for T cells than macrophages, and even so-called macrophage-tropic viral clones were ~10-fold less infectious for macrophages than T cells [15, 16].”

The authors should include the recent findings by Ganor and colleagues (Nature Microbiology, PMID: 30718846) reporting on the persistence of infected macrophages in urethral macrophages from patients under ART. These also displayed VCCs, further supporting a role for these structures in viral spreading. In this context, the authors could speculate a possible role for residual viral transmission via macrophages as a mechanism contributing for preservation of the viral reservoir under ART. 

We were aware of this and other studies that support a role for macrophages as tissue reservoirs in vivo, however we chose not to include them in the review since these studies do not provide evidence for cell-cell spread, which is the focus of the current review, and since other reviews in this series will address macrophages as viral reservoirs. Nevertheless, we now mention the Ganor study in the context of VCC (line 145) as follows. “Adding in vivo relevance to the idea of HIV-1-containing VCC, Ganor and colleagues reported that macrophages isolated from urethral tissue of HIV-1-infected men under cART contained HIV-1 DNA, RNA, protein and virions in a VCC-like compartment. By contrast, viral components were undetectable in urethral T cells, highlighting the potential importance of macrophages as viral reservoirs in specific mucosal tissues [52].”

In the section devoted to the VCC and its connection to the plasma membrane, the authors should quote here two additional studies. In the first one, 10.1371/journal.pone.0069450, evidence supporting immediate access of the VCC to small dextran present in the extracellular medium was presented. In addition, the data presented revealed the dynamic nature of the VCC connection to the plasma membrane. In the second one, 10.1084/jem.20130566, time lapse microscopy of the same macrophages before and after infection demonstrated that HIV-1 can subvert pre-existing CD36+ compartments and convert them into VCCs.

We thank the reviewer for this suggestion and have now added the paragraph below. “Live-cell imaging revealed that some VCC were open to the extra-cellular milieu and could be accessed by small dextran particles after HIV-1 infection. However not all VCC remained cell surface-connected, some conduits being transient and allowing or not the diffusion of small molecules in and/or out these compartments [47]. The origins of VCC remain unclear, as this structure is pleomorphic [42] and presents markers specific for the plasma membrane (eg. CD44), but also of MVB (eg. CD9, CD81, CD53) [41][39]. It was subsequently established that HIV-1 subverts a pre-existing compartment in macrophages to form the VCC. Following infection of MDM, viral Gag is recruited to pre-existing CD36+ compartments which become VCC [42, 48].

Minor issues:

Line 101: While the authors refer that HIV virions in VCCs remain infectious for only a few days, the reference they cite refers a period of 6 weeks within which infectious virions could be recovered. They authors could here point to the limitation of the measurement of infectious viral particles performed in this study.

We appreciate this suggestion and have added the paragraph below to address it. However, we are not sure what the limitation of measurement of infectious virus particles is that the reviewer refers to in this study, and would be grateful for guidance from the reviewer in this respect. “However, subsequent analysis revealed that these vesicular structures were in fact surface-connected and at neutral pH [40-42], and therefore were a non-degradative compartment within which HIV-1 could remain infectious for extended periods of time. This was proposed to be up to 7 weeks post-infection in one study [43], although this length of time might be confounded by very low level ongoing viral replication in the presence of inhibitor.”

It would be interesting to include the findings by Hammonds and colleagues (Plos Pathogens, 2017, PMID 28129379) when discussing in cis and in trans infection, or, alternatively, in the VCC section. This work reports that productive infection of macrophages is not required for formation of the VCC and that virions that result from productive macrophage infection can intermingle with virions captured from the extracellular space via SIGLEC-1.

We agree and have added the paragraph below in this respect. “Additionally, lectins such as Siglec-1 can play an important role in VCC formation and function. A recent study demonstrated that VCC formation does not necessarily require macrophage infection, since Siglec-1 capture of non-infectious viral-like particles (VLP) bearing HIV-Env and gangliosides also led to VCC formation [49]. Depletion of Siglec-1 from infected macrophages decreased VLP uptake, significantly reduced VCC volume and reduced the transfer of particles to autologous T cells, confirming that VCC are important structures for viral transfer between infected macrophages and T cells [49].”

General author comment

We also added an additional reference we became aware of and description to the “Fusion between HIV-1-infected CD4+ T cells and macrophages” section, line 227 as follows: “Env-mediated fusion between monocytic and T cell lines revealed that the ensuing heterokaryons were viable, stable, and presented a dominant activated monocyte-like phenotype [84].

Reviewer 2 Report

This is a detailed review that focuses on macrophages and cell transmission. It extensively reviews the cell-to-cell interactions and pathways that can lead to macrophage infection. The review is well written. Criticisms are minor and are outlined below.

I would suggest modifying the title to reflect the focus on maybe something like Macrophages and cell-cell interactions: an HIV infection hub.

There is limited discussion of macrophage function and subsets and how these may engage T cells and other myeloid cells by different mechanisms (this concept is raised almost as an afterthought in the discussion).

There is an opportunity to discuss the cell-cell synapses in greater detail, the molecules and if they signal or have any functional relevance beyond holding cells together. 

Some of these mechanisms have been contentious and there would be added value to the field to discuss some of the more contradictory literature to provide a balance view. 

Author Response

Comments and Suggestions for Authors – REVIEWER 2

Author replies are in in italics for clarity.

This is a detailed review that focuses on macrophages and cell transmission. It extensively reviews the cell-to-cell interactions and pathways that can lead to macrophage infection. The review is well written. Criticisms are minor and are outlined below.

We thank this reviewer for their supportive comments and we address their comments and criticisms below.

I would suggest modifying the title to reflect the focus on maybe something like Macrophages and cell-cell interactions: an HIV infection hub.

We thank the reviewer for this suggestion, and in the end decided to go with: “Macrophage cell-to-cell interactions promoting HIV-1 infection”

There is limited discussion of macrophage function and subsets and how these may engage T cells and other myeloid cells by different mechanisms (this concept is raised almost as an afterthought in the discussion).

We agree that macrophage polarization and cell-cell HIV-1 spread were not discussed, and we now include the following paragraph in the section CD4+ T cell infection by spread from macrophages (line 177). “Macrophage phenotype is highly influenced by cytokine environment, and differentiation of MDM into ‘classically activated’ (M1) or ‘alternatively activated’ (M2a, M2b, M2c) MDM has a profound influence on sensitivity to HIV-1 infection [62]. Moreover, macrophage phenotype has implications for function in terms of HIV-1 cell-cell spread. MDM differentiated in the presence of IL-4 (termed M-4) were found to be more sensitive to HIV-1 infection and to subsequent dissemination of HIV-1 to T cells than MDM differentiated in IL-13 (M-13) [63]. MDM differentiated to M1 in the presence of IFN-g and TNF-a, or M2a in the presence of IL-4, reduced or enhanced expression of DC-SIGN respectively, which in turn modulated sensitivity to HIV-1 infection and dissemination to T cells [64], highlighting the role of DC-SIGN and related adhesion factors in HIV-1 spread. A recent analysis of macrophage phenotype in ex vivo urethral tissues [52] revealed a switch from a predominantly M1 phenotype in uninfected individuals to an intermediate (Mi) in infected, ART-suppressed individuals, implying that Mi polarization may facilitate HIV-1 reservoir formation.”

There is an opportunity to discuss the cell-cell synapses in greater detail, the molecules and if they signal or have any functional relevance beyond holding cells together. Some of these mechanisms have been contentious and there would be added value to the field to discuss some of the more contradictory literature to provide a balance view.

This is an interesting suggestion, and we would be pleased to address any controversial ideas in the field. However, we are confined within this collection of reviews to a focus on cell-cell spread of HIV-1 to and from macrophages. Within these limits, and beyond what we have added above concerning macrophage polarization and cell-cell spread, we are unsure which literature the reviewer refers to. If they could perhaps specify we would be pleased to include additional material. 

General comment

We also added an additional reference we became aware of and description to the “Fusion between HIV-1-infected CD4+ T cells and macrophages” section, line 227: “Env-mediated fusion between monocytic and T cell lines revealed that the ensuing heterokaryons were viable, stable, and presented a dominant activated monocyte-like phenotype [84].